# Ethnic Variations in Nutritional Status among Preschool Children in Northern Vietnam: A Cross-Sectional Study

**DOI:** 10.3390/ijerph16214060

**Published:** 2019-10-23

**Authors:** Thi Tuyet Le, Thi Thuy Dung Le, Nam Khanh Do, V. Savvina Nadezhda, M. Grjibovski Andrej, Thi Trung Thu Nguyen, Thi Thanh Mai Nguyen, Thi Tuyen Vu, Thi Huong Le, Thi Thu Lieu Nguyen, Thi Anh Dao Duong

**Affiliations:** 1Department of Human and Animal Physiology, Faculty of Biology, Hanoi National University of Education, 136 Xuan Thuy Street, Hanoi 100000, Vietnam; trungthu@hnue.edu.vn (T.T.T.N.); daodangduc@gmail.com (T.A.D.D.); 2Department of Pediatrics, Hanoi Medical University, 1 Ton That Tung Street, Dongda District, Hanoi 100000, Vietnam; letono2002@gmail.com (T.T.D.L.); mainguyenhmu@gmail.com (T.T.M.N.); 3Department of Public Health, Healthcare, General Hygiene and Bioethics, North-Eastern Federal University, 677000 Yakutsk, Russia; andrej.grjibovski@gmail.com; 4Department of Nutrition and Food Safety, Institute of Preventive Medicine and Public Health, Hanoi Medical University, 1 Ton That Tung Street, Dongda District, Hanoi 100000, Vietnam; donamkhanh@hmu.edu.vn (N.K.D.); vuthituyen1596@gmail.com (T.T.V.); lethihuong@hmu.edu.vn (T.H.L.); lieu.nguyen1508@gmail.com (T.T.L.N.); 5Central Scientific Research Laboratory, Northern State Medical University, 163000 Arkhangelsk, Russia; nadvsavvina@mail.ru; 6Department of Health Policy and Organization, Al-Farabi Kazakh National University, Almaty 050040, Kazakhstan; 7West Kazakhstan Marat Ospanov Medical University, Aktobe 030019, Kazakhstan

**Keywords:** nutritional status, preschool children, minor ethnicity, Vietnam

## Abstract

(1) Background: Vietnam is a multiethnic country undergoing rapid economic development, the improvement in nutritional status in preschool children is not equally shared by all ethnic groups; (2) Methods: A cross-sectional study was performed from September–December 2018 on 16,177 children aged 24–60 months representing Kinh (*n* = 14421), Muong (*n* = 1307) and Tay (*n* = 449) ethnic groups. Prevalence of malnutrition, overweight, obesity and anthropometric indices were compared across ethnic groups, using WHO 2006 child growth standards; (3) Results: The prevalence of malnutrition among children of Kinh, Muong and Tay origins was 14.7%, 34.3% and 43.2%, respectively. The corresponding data for overweight was 5.5%, 2.7%, 2.2% and for obesity 2.8%, 0.8% and 0.4%, respectively. The prevalence of stunting remained the highest in three subtypes of malnutrition in all ethnic groups. Kinh children were heavier and taller than the other groups, while Muong children were taller than Tay children (*p*-value < 0.001); and (4) Conclusions: Malnutrition remains a major public health issue among children of minor ethnicities while overweight and obesity is an emerging challenge for the Kinh ethnic group. The results imply that a community-based intervention should be specific to ethnicity to reduce the gap in nutritional status between ethnic groups in Vietnam.

## 1. Introduction

Appropriate nutrition has a distinct relationship with comprehensive development of children [1]. Deviation from normal nutritional status described as being underweight, overweight, or obese can lead to health issues for children under five [2,3]. The double burden of malnutrition and obesity in children across developing countries represents a significant challenge. As reported by World Health Organization (WHO) in 2014, 42 million children aged under five were overweight or obese and greater number was stunting (156 million) and wasting (50 million) [4]. The prevalence of overweight and obese children has increased to an alarming level in developing countries. For example, being overweight and obesity have been reported in children in China (9.5%), Indonesia (5.1%), Kyrgyz Republic (28.2%) [5]. The prevalence of being overweight and obese is known to be associated with economic development and urbanization [6]. The World Bank reports the economy of Vietnam has made a significant economic development over the last two decades [7]. Along with economic growth, the change in lifestyle and nutritional transitions [8] have contributed considerably to prevalence of overweight children and childhood obesity. This is particularly evident in the most urbanized areas of Vietnam. A Vietnamese study conducted in Ho Chi Minh City (the financial center of the country) in 2007 estimated the prevalence of being overweight or obese to be 20.5% and 16.3% [9]. A study in Hanoi in 2015 also revealed the prevalence of overweight children and childhood obesity combined was greater in urban areas (21.1%) than in rural areas (7.6%) [10]. Overweight children and childhood obesity are more likely to lead to adult obesity associated with increased mortality and morbidity [10], therefore this issue requires close attention of public health professionals.

Malnutrition also continues to be a challenge for Vietnam. According to the official data in 2014, under-five mortality was 20.9 per 1000 with half of these deaths related to malnutrition [11]. The proportion of malnourished preschool children is still high in Vietnam, especially in rural and mountain areas. In the Quanba district of Hagiang—one of the poorest areas of the country—populated by ethnic minorities—the prevalence of underweight, stunting and wasting among under five children was 24.8%, 77.3% and 4.5% in 2016, respectively [12]. Another study in rural Northern district of children aged 12 to 36 months indicated a prevalence of being underweight (7.6%), stunting (23.5%), wasting (6.7%) with anemia and vitamin D deficiency [13]. A longitudinal study in children aged 10–60 months in the Northern provinces and in Hanoi revealed that 15.7% children were stunted, 4.3% children were underweight and 3.3% were both stunted and underweight [14]. Even in urban areas of Ho Chi Minh, the prevalence of stunting was 8.2% [15]. However, prevalence of malnourishment exists predominantly among the ethnic groups who live outside the urban areas.

Vietnam is a country with 54 ethnic groups. The Kinh ethnic group comprises 85.3% of the population [16]. In Northern Vietnam, ethnic minorities mostly live in mountains, midlands or rural parts of the country with less privileged living conditions and educational opportunities compared to the Kinh. The United Nations International Children’s Emergency Fund Child poverty reported the Kinh consisted 14.1% while for other ethnic groups, this proportion was 52.4%, (UNICEF) [17]. The educational and economic gap that exists between Kinh and the other ethnic minorities also lead to the ethnic differences in nutritional status of children under five. Previous studies on nutritional status among preschool children were mainly performed on the Kinh ethnic group in Hanoi and Ho Chi Minh City. Therefore, evidence on nutritional status of ethnic minorities in Vietnam is very limited. In this research, we aimed to compare main indicators of nutritional status in the three ethnic groups (Kinh, Muong, Tay) using the WHO data as a reference population for standardization [18] to clarify the ethnic distinctions in nutritional status among children under five.

## 2. Materials and Methods

The sample was collected from September to December 2018 in seven provinces of Northern Vietnam. Participants took part in a population-based cross-sectional study using cluster sampling. Sample size for a cross-sectional study was calculated using Epi Info software [19]. A sample size of 402 in the smallest group is sufficient to produce a two-sided 95% exact confidence interval with a precision of 5% for the expected prevalence of malnutrition of 50%. The sample included Kinh as the major ethnic group and other minor ethnicities (Muong and Tay), considering their popularity in Northern Vietnam. In the Kinh group, children were taken from randomly chosen kindergartens of Hanoi (12 suburban schools (*n* = 4048), 24 urban schools (*n* = 7734)) and Thanhhoa, Namdinh, Phutho provinces (each province with 2 schools (*n* = 1916)). The minor ethnicities (Muong and Tay) were randomly selected with consideration to their primary living areas, Hoabinh province (Kimboi district) for Muong group and the Caobang province (Quanguyen district) and Backan province (Nganson district) for the Tay group. These children were invited to primary healthcare settings for a health check (visit proportion >90%) and being involved into the research. Children of appropriate age with parental permission to be involved into research were included, while children who were overweight/obese or malnourished because of medical reasons and those whose parents did not sign informed consent were not included in the study. In total, 16,177 children aged 24–60 months old were recruited in this study with the smallest group (Tay, *n* = 449) ensuring sufficient sample size for calculating confidence intervals for the estimates with the selected level of precision.

Individual height and weight measurements were taken twice by trained social workers and the average value was used for the study. All measurements were taken with subjects wearing lightweight clothing and without shoes. The BMI was calculated as the weight per square kilogram of height (kg/m^2^). The height-for-age, weight-for-age, weight-for-height-for-age, and BMI-for-age Z-scores were calculated using the WHO 2006 reference values. Overweight was defined as weight-for-age or weight-for-height-for-age or BMI-for-age Z-scores between 2 and 3; these Z-scores of 3 or above indicated obesity; malnutrition was defined as weight-for-age, weight-for-height-for-age or BMI-for-age Z-scores lower than −2 or its combination. Stunting was determined as height-for-age Z-scores to be below −2. Underweight children were those with weight-for-age Z-scores below −2 while wasting-weight-for-height-for-age Z-scores below −2. Children who were overweight/obese or malnourished because of medical reasons and those whose parents did not sign informed consent were not included in the study.

Prevalence of overweight, obesity, malnutrition with subtypes underweight, stunting and wasting across the three ethnic groups was calculated with 95% confident intervals, using the method of Newcombe, OR was identified of various nutritional status of Muong and Tay ethnic groups in compare to Kinh group via univariate regression. Numeric data (Z-score values, the mean of height and weight of boys and girls) were analysed and compared by using one-way ANOVA and Tukey’s post-hoc test. All calculations were performed using SPSS version 16.0 (SPSS, Chicago, USA) or Stata 15 software (Stata Corp LLC, TX, USA). *p* < 0.05 is considered as statistically significant. The procedure and protocol of research were approved by the ethnics committee of the Vietnam National Institute of Nutrition (Protocol No. 343/VDD-QLKH dated 27/7/2018). Informed consent agreement was obtained from each participant’s parent.

## 3. Results

In total, 16,177 children aged 24–60 months were involved in our study. The sample consisted predominantly of boys (*n* = 8568; 53%). The boy dominance was also reflected in each ethnicity with the Kinh group (*n* = 14,421, 53.1% boys), Muong group (*n* = 1307, 51.6% boys) and Tay group (*n* = 449, 53% boys).

### 3.1. Prevalence of Various Nutritional Statuses in Our Sample

Calculated in both genders, the prevalence of malnutrition was Kinh (14.7%), Muong (34.4%), Tay (43.2%). The statistics for being overweight revealed 5.5% of the sample with, 2.7% (Kinh), 2.2% (Muong) and 2.8% (Tay) and in the same order for obesity: 2.8%, 0.8% and 0.4%. The difference in malnutrition with three subtypes (underweight, stunting, wasting), overweight and obesity in each gender of three ethnic groups is illustrated in Table 1.

The proportion of malnutrition in Kinh group was remarkably lower than Muong and Tay groups in both genders. As results of univariate regression, malnutrition prevalence of Muong and Tay was 3.5 times higher (95% CI = 3.18–3.82) and 4.6 (95% CI = 3.85–5.56) than Kinh children. Comparison between the Muong and Tay minor ethnicities highlighted malnutrition prevalence of Muong was lower than Tay who demonstrated malnutrition only in boys. For three subtypes of malnutrition across all ethnic groups, in both genders the descending order of prevalence was stunting, underweight, wasting, respectively. Prevalence of stunting was registered highest in Tay group, followed by Muong group, and finally, Kinh group. While the underweight prevalence remains same higher than the Kinh group in Muong and Tay groups. The difference in wasting prevalence in both boys and girls was noted only between Kinh and Muong groups. Proportion of overweight and obesity in Kinh group boys was highest. The proportion of overweight and obesity in girls of Kinh group was higher in the Muong, not the Tay group. There was almost no gender difference in nutritional status of all three ethnic groups, except for the proportion of overweight and obesity in Kinh boys, which was higher than that of the girls in the Kinh group. However, prevalence of overweight and obesity combined across three ethnic groups, Muong and Tay ethnic groups showed the inferior in compare to Kinh group with OR = 0.36 (95% CI = 0.28–0.46) and OR = 0.18 (95%CI = 0.08–0.4), respectively.

### 3.2. Comparison of the Main Anthropometric Indices of Preschool Children across Three Ethnicities

In comparing the nutritional status of children in three ethnic groups, it is obligatory to provide values of the main anthropometric indices of each ethnicity, compared by one-way ANOVA.

Table 2 shows the differences in the mean values of weight and height and other anthropometric indices of three ethnic groups. All three ethnic groups had negative values of weight-for-age, height-for-age Z-score; the Z-score of Kinh was higher than Muong, which was higher than Tay group. For the Z-score of BMI and weight-for-height-for-age, the values were positive in the Kinh group, in contrast with negative values of those of Muong and Tay groups. In consideration of P-values, we could see the significant distinction in mean values of weight and height and height-for-age Z-score of children in three ethnic groups in order from Kinh, then Muong, then Tay. For weight-for-age and weight-for-height-for-age and BMI Z-scores, there was no difference between the Muong and Tay groups, but both of them were inferior in comparison to Kinh children. Figure 1 and Figure 2 were created for more detailed comparison.

Notable difference of distribution was noted between Kinh group and minor ethnicities (Muong, Tay) in length-for-age Z-score and weight-for-age Z-score, the left shift of Muong and Tay than Kinh group and the last also was left sided in compare with WHO reference. The curves of weight-for-length Z-score and BMI-for-age Z-score of Kinh, Muong, Tay and WHO reference were similar.

Figure 2 illustrates the difference of growth in height and weight by age in boys and girls. The similarity of ethnic Tay and Muong was documented while the increase in the Kinh group was obvious. All three lines of data were located lower than the standards of WHO. The growth chart of weight and height of Kinh ethnic in both genders appeared more even than those lines of Muong and Tay, which showed greater variation. The chart of Muong and Tay groups demonstrated greater separation from the Kinh group (and reference data) over the age of the child.

## 4. Discussion

There were 16,177 children aged 24–60 months included in this study, with Kinh children representing the largest proportion of the sample, which was reflective of the Vietnamese population. Muong and Tay children were chosen because they are one of the largest minor ethnicities, whose habitats are located in several provinces of Northern Vietnam [20]. Consequently, our sample could be stated to be representative for preschool children of Northern Vietnam. In Vietnam, gender imbalance has been recorded for many years. In 2009, Christopher reported the gender ratio at birth (110 boys:100 girls) [21], in 2016 the Government declared in Delta of River Hong (1137 boys:100 girls) [22]. In our study, male gender dominance was demonstrated Kinh group (*p* < 0.05, Chi-square test), for Muong and Tay groups, the *p* value was above 0.05. The gender difference is considered to be a matter of choice for many parents who practice gender identification in utero and selective abortion with the primary aim of a male birth, especially in urban and suburban areas of the Kinh group, despite its illegality in Vietnam. For minor ethnicities, the sex selection of fetus and abortion was not as easy as just delivery, as a result the dominance of boys than girls was insignificant in these ethnicities. However, the sex imbalance was noted in Table 1 but gender prevalence was not evident in nutritional status between boys and girls in each ethnicity, except for the proportion of overweight and obese boys was higher than girls in Kinh group, but this proportion was also quite low (<10%). Malnutrition and all subtypes (stunting, wasting and underweight) was almost the same for girls and boys in three ethnicities. The situation of India in 2009 was recorded differently with the tendency for malnourished prevalence in girls [23], or in Iran, stunting boys were 1.41 times higher than girls [24]—that might relate to the cultural and religious practices of these countries. In Vietnam, people do not engage in gender discrimination in terms of childcare, especially in preschool ages. All children above the age of 2 in the family are generally fed with family’s food, irrespective of gender. As such, sex distinction in nutritional status in childhood in Vietnam is rarely found.

The results of Table 1 show a double burden in nutritional status in Kinh children with the proportion of malnutrition in boys (13.9%), girls (15.9%) and over-nutrition (overweight and obesity) in boys (10.3%), girls (5.9%). For other ethnicities, malnutrition was the main problem; prevalence of malnutrition was much higher than over-nutrition. Prevalence of malnutrition was higher than obesity in all three ethnicities. The dominance of malnourished children over obese children indicates the lack of nutrition in Vietnam in general. Vietnam has experienced a long history of wars and its consequences resulting in more than 2 million children under five being malnourished [25].

Prevalence of underweight children and stunting in the 1997–1998 survey was 33.1% and 66.9% [26]; this proportion was slightly decreased until 2008 with 31.8% of children being underweight and 44.3% [27] of stunting. The newest data from 2019 showed the impressed reduction to underweight 7.6% and stunting of 23.5% in a rural province [13], but in general, nowadays the prevalence of stunting children of Vietnam still in high group among Southeast Asian countries [28]. Proportion of malnutrition in Tay children was higher than that of Muong children. Both two ethnicities were higher than Kinh children with OR = 3.5 (95%CI = 3.18–3.82) for Muong and 4.6 (95%CI = 3.85–5.56) for Tay, respectively. Children with who are overweight and obese were remarkable in Kinh group but insignificant in both Tay and Muong groups. A certain percentage of over-nutrition in Kinh children also raises the issue of nutrition that a developing country like Vietnam faces. The malnutrition prevalence in Kinh children resembles that of under five year old children of poor areas of China (14.3%) in 2016, while the prevalence of obesity and overweight was lower (5.5% overweight and obesity 2.8% in comparison with a combination of 21.4% in Chinese children) [29,30]. Overweight and obesity were more prevalent in urban children while the higher prevalence of children malnutrition was contributed in rural and mountainous areas. This is understandable because economically advantaged urban families have greater access to food choices, but are time-poor and tend toward fast food, children have reduced physical activities due to in-house entertainment such as watching TV and using smartphones. In contrast, mountainous and rural areas live in poor economic conditions, families often have greater numbers of children, children are active in family work life, access to and choice of food sources is poor, therefore the rate of malnourished children is much higher. This difference reflects findings from Pakistan with the highest percentage of underweight children in rural areas (64.7%) and overweight in urban areas (9.4%) [31]. In Iran, which demonstrated significant difference in BMI of 19,824 children under five with the rate of overweight and obese children in urban areas at 5.3% and 1.6% in comparison with rural areas, at 2.7% and 0.6%, respectively [32]. The data revealed the nutritional status of Kinh children and the prevalence of overweight children and childhood obesity was lower than the findings in a study on 670 preschool children of Ho Chi Minh City (20.5% and 16.3%, respectively) [9]. The differences are explained in that the sample for the Kinh ethnic group was taken from various provinces, but one area of the country and our sample was larger in numbers and representative for Northern Vietnam.

Malnutrition of children is an indicator of poverty of a country [33]. This is demonstrated by the malnutrition prevalence in Muong and Tay children from the minor ethnic groups as compared to the more socially and economically advantaged Kinh group. The World Bank Group reported in 2016 that the Kinh and Hoa people (originated from China, a small ethnicity, but living in market-centre areas of Ho Chi Minh City), has the poverty headcount rate only 3.2%, much lower than minor ethnicities (44.6%) [34]. It could be suggested that the nutritional status of Kinh children with a double burden (under and over-nutrition) is typical of a rapidly developing country with a middle-income population, whilst dominance of malnutrition in Muong and Tay more closely resembles low income regions. Several worldwide summaries reported malnutrition prevalence of under-five-year-old children of poor areas of China (14.3%) in 2016, while the this study’s findings of obesity are much lower (5.6% in comparison with 21.4%) [29,30]. Counting malnutrition and its subtypes, it was noted that prevalence of malnutrition most resembled the prevalence of stunting, not underweight or wasting, in all three ethnicities. Stunting children or children in moderate or extreme poverty, who are at risk of poor development, as a study on 141 countries in 2010 [35] or malnutrition of rural poor regions of India (24.4%) [36]. Stunting prevalence also indicated chronic malnutrition. The problem of malnutrition was more serious in Muong and Tay groups than Kinh children. The economic gap of different ethnicities affects the structure of nutritional status of preschool children.

Vietnam is experiencing a widespread and rapid urbanization, along with a rapid change of economy, especially in biggest Northern cities—the site of the Kinh group. The speed of modernization in the minor ethnicities—rural, mountainous and midlands areas—is much slower. Therefore local residents’ lifestyles and socioeconomic status are changing with a slower speed. For the Kinh children, percentage of stunting was found the same as that of children under 5 years old in Thailand (14.1%) [37], while Muong and Tay groups, found similarity with Indonesia (approximate 37%) [38]. This stunting prevalence of children in developed countries is low, according to data from UNICEF published in 2018; for example, less than 2.5% in Australia and New Zealand, or 2.6% in USA [39]. In a study undertaken by Huong L.T at a mountainous, poor district with majority population consisted by minority ethnics, the result on sample size showed highest prevalence in a year of underweight (24.9%), stunting (29.8%) and wasting (14.3%) [25]. This supports the findings that the problem of malnutrition in preschool children remains challenging with the minority ethnic groups of Vietnam.

Considering anthropometric indices across three ethnic groups, though the mean of weight and height of Kinh group was greater than that of Muong group, the lowest was Tay group. In general, the weight-for-age, weight-for-height-for-age and BMI Z-scores of two minor ethnicities were not statistically significant, but they were lower than that of Kinh group (*p* < 0.05). There is a notable gap between Muong and Tay as minor ethnic groups than Kinh children. For height-for-age Z-score, we also found the descending order: Kinh, Muong, Tay. The Muong ethnic groups live near Hoabinh, a province not too far from the center, Hanoi (about 100 km) in comparison to the Tay ethnic (more than 300 km), it may also effect on nutritional status of children. Comparing the mean of all Z-score, it was almost negative values, our children is still lower than WHO population of reference. On the other hand, the weight-for-age and height-for-age Z-score distribution was shown the left shift of Vietnamese children in compare with WHO reference population, the median value of Z-score of all three ethnicities was negative, the left shifted curves (Figure 1) claimed that the issue of malnutrition was more severe than the problem of overweight and obesity in children under 5 years old. Our situation resembles the distribution curve of Indian children under 3 years old since 2005 [40]. The weight-for-age and height-for-age Z-score of Kinh was rightly located in compared to Muong and Tay, two resemble each other curves, and for more detailed. Figure 2 demonstrated the mean of height and weight in each gender of minor ethnic was much lower than Kinh nation. Even the lines of mean values of Kinh nation have several points approaching to the WHO lines. The noted apparent inferiority in height and weight of Muong and Tay children compared to Kinh children indicated that adequate nutrition for children seems to play an important role in the nutritional status of children aged below five.

This data will serve as background information to promote a long-term strategy, especially reserved for children of minor ethnicities to improve childhood nutritional status nationally. For the high prevalence of malnutrition, especial stunting in children of minor ethnicities, local health providers should pay focus on parental education; pre-school children will benefit from nutritious food, local food resources, and formula milk to improve their underweight and stunted conditions. Educational administrators should encourage parents to participate in school nutrition programs [41] to gradually improve anthropometric indices of minor ethnic children.

In our study, we evaluated the nutritional status of children of minor ethnicities (Muong and Tay) and compared it with the Kinh ethnic group, representing the majority in Vietnam and with references from the WHO. Our study also has material limitations in that while indicating the differences between childhood nutritional status, related factors to nutritional status—such as educational, economic, cultural factors, etc.—were not covered. To clarify the social–environmental impacts on nutritional status of children between various ethnic groups, further investigation is required.

## 5. Conclusions

The prevalence of malnutrition and stunting in preschool children in a developing country like Vietnam has been a problem in regards to the nutrition status of children. While malnutrition is particularly serious in two ethnic groups, Muong and Tay (considered minor ethnicities), overweight and obese children were more prevalent in the economically stable Kinh ethnic group. The difference in the distribution of nutritional status among children under 5 from these ethnic groups was statistically significant. Weight and height by age of ethnic minority children is lower than that of Kinh children. The height of Vietnamese children is generally lower than the WHO standards. Further positive measures are needed to manage and support child nutrition in order to fill the gap between children of minor ethnicities and ethnic Kinh to address the comprehensive development of children to help ensure a stable future for the Vietnamese society.

## Figures and Tables

**Figure 1 ijerph-16-04060-f001:**
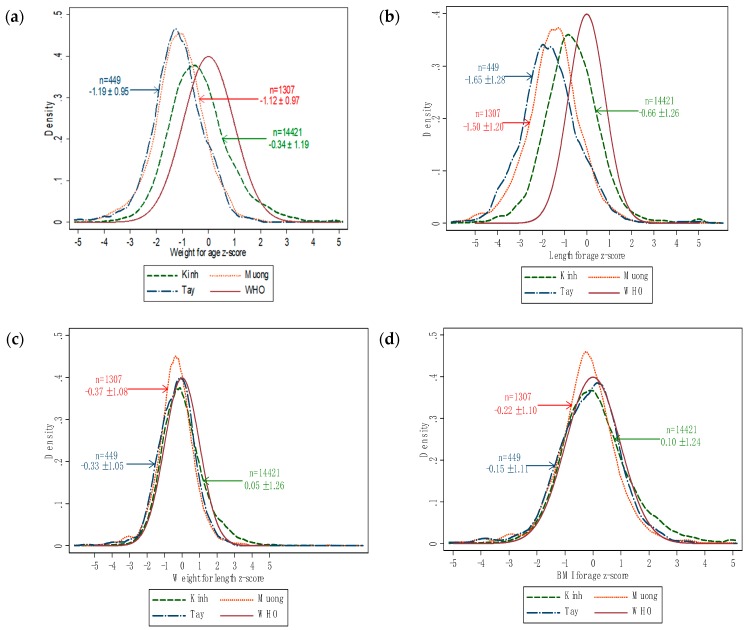
Distribution of weight-for-age (**a**), length/height-for-age (**b**), weight-for-length-for-age (**c**) and BMI-for-age (**d**) Z-scores across ethnic groups in comparison with WHO reference. (“*n*” is number of children in each ethnic group).

**Figure 2 ijerph-16-04060-f002:**
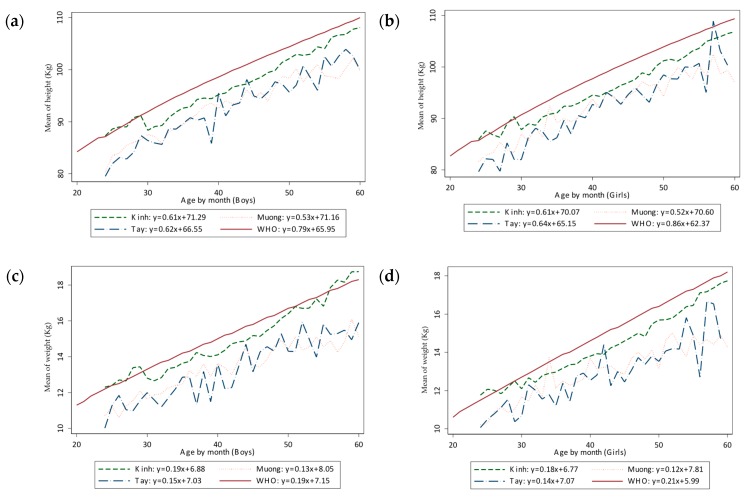
The development of height and weight by age in different ethnicities in comparison with WHO 2006 reference population. (**a**) height in boys; (**b**) height in girls; (**c**) weight in boys; (**d**) weight in girls.

**Table 1 ijerph-16-04060-t001:** Prevalence of various nutritional statuses across ethnic groups among 24–60 month old children in Northern Vietnam.

Nutritional Status	Kinh (*n* = 14,421; 89.1%)		Muong (*n* = 1307; 8.1%)		Tay (*n* = 449; 2.8%)	
	Prevalence (%)	95% CI (%)	Prevalence (%)	95% CI (%)	Prevalence (%)	95% CI (%)
**Boys**						
Malnutrition	13.9	13.1–14.8	35.1	31.6–38.8	48.2	41.9–54.7
Underweight	*5.3*	*4.8*–*5.8*	*16.7*	*14.1*–*19.7*	*17.8*	*13.4*–*23.3*
Stunting	*11.5*	*10.9*–*12.3*	*31.9*	*28.5*–*35.5*	*43.9*	*37.7*–*50.4*
Wasting	*3.1*	*2.7*–*3.5*	*6.5*	*4.9*–*8.6*	*5.2*	*3.0*–*8.9*
Overweight	6.5	6.0–7.1	1.5	0.8–2.7	2.2	0.9–5.0
Obesity	3.8	3.5–4.3	1.2	0.6–2.3	0.0	0–1.6
**Girls**						
Malnutrition	15.6	14.7–16.5	33.4	29.8–37.2	37.9	31.7–44.5
Underweight	*6.1*	*5.6*–*6.7*	*13.8*	*11.3*–*16.7*	*15.5*	*11.3*–*20.9*
Stunting	*12.6*	*11.8*–*13.4*	*29*	*25.6*–*32.1*	*37.4*	*31.3*–*44.0*
Wasting	*3.1*	*2.7*–*3.6*	*5.2*	*3.7*–*7.2*	*3.2*	*1.6*–*6.5*
Overweight	4.3	3.9–4.8	1.9	1.1–3.2	2.3	1.0–5.2
Obesity	1.6	1.3–1.9	0.3	0.1–1.2	0.9	0.3–3.6

95% CI was calculated by methods of Newcombe.

**Table 2 ijerph-16-04060-t002:** Main anthropometric indices of three ethnic groups.

Anthropometric Index	Kinh (1)		Muong (2)		Tay (3)		*p*_1_–_2_	*p*_1_–_3_	*p*_2_–_3_
	Mean	95% CI	Mean	95% CI	Mean	95% CI			
Weight (kg)	15.5	9.1; 21.9	13.2	9.0; 17.4	12.6	8.4; 16.8	*<0.0001*	*<0.0001*	*0.001*
Height (cm)	99.2	83.2; 115.2	92.9	77.4; 108.3	90.5	74.1; 106.9	*<0.0001*	*<0.0001*	*<0.0001*
Weight-for-age Z-score	−0.3	−2.7; 2.0	−1.1	−3.1; 0.8	−1.2	−3.1; 0.7	*<0.0001*	*<0.0001*	0.286
Height-for-age Z-score	−0.7	−3.2; 1.9	−1.5	−3.9; 0.9	−1.7	−4.2; 0.7	*<0.0001*	*<0.0001*	*0.038*
Weight-for-height-for-age Z-score	0.05	−2.4; 2.6	−0.37	−2.5; 1.8	−0.3	−2,3; 1.8	*<0.0001*	*<0.0001*	0.654
BMI-for-age Z-score	0.10	−2.4; 2.6	−0.2	−2.4; 2.0	−0.2	−2,4; 2.1	*<0.0001*	*<0.0001*	0.255

*p* value received from one-way ANOVA test via Tukey’s post-hoc test. Comparison between Kinh and Muong (*p*_1-2_), between Kinh and Tay (*p*_1-3_), between Muong and Tay (*p*_2-3_).

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
