# Peer review of "Ethnic Variations in Nutritional Status among Preschool Children in Northern Vietnam: A Cross-Sectional Study"

_ijerph, 2019, doi:10.3390/ijerph16214060_

Round 1
Reviewer 1 Report
The article sent to me for review is very interesting and addresses important topics. However, some information should be clarified or added:
Material and methods: Authors should add information about calculation of sample size, inclusion criteria and flow chart of study participants Results: In table 2 please add a value of P (not only 1 or 2), please add OR of incidence of nutritional status in relation to Ethnic Discussion: line 169-174 should be in "Material' Limitation section should be added What could have influenced such results? Did the authors take into account disturbing factors such as parents' education, socioeconomic status, number of children in the family, etc. (about which the authors write in the discussion)
Reviewer 2 Report
It is a decent manuscript with significant interest to Vietnamese population and health care providers. If the authors could document/comment on the shift of malnutrition from the last conducted study in the results section, that would enhance the overall strength of the paper by indicating the development of the country in the last decade.
Additionally, in the discussion, I would encourage the authors to discuss about some local products/measures that the healthcare practioners should follow to combat the issue of malnutrition in the country among the various ethnic groups.
Round 2
Reviewer 1 Report
The authors have made all corrections recommended by the reviewer. The work deserves publication in the journal.